# Effects of Prolonged Medical Fasting during an Inpatient, Multimodal, Nature-Based Treatment on Pain, Physical Function, and Psychometric Parameters in Patients with Fibromyalgia: An Observational Study

**DOI:** 10.3390/nu16071059

**Published:** 2024-04-04

**Authors:** Daniela A. Koppold, Farid I. Kandil, Anna Müller, Oliver Güttler, Nico Steckhan, Sara Meiss, Carolin Breinlinger, Esther Nelle, Anika Rajput Khokhar, Michael Jeitler, Etienne Hanslian, Jan Moritz Fischer, Andreas Michalsen, Christian S. Kessler

**Affiliations:** 1Institute of Social Medicine, Epidemiology and Health Economics, Charité—Universitätsmedizin Berlin, Corporate Member of Freie Universität Berlin and Humboldt-Universität zu Berlin, 10117 Berlin, Germanyanna.mueller@charite.de (A.M.); nico.steckhan@charite.de (N.S.); sara.meiss@charite.de (S.M.); carolin.breinlinger@charite.de (C.B.); jan-moritz.fischer@charite.de (J.M.F.);; 2Department of Internal Medicine and Nature-Based Therapies, Immanuel Hospital Berlin, 14109 Berlin, Germany; 3Department of Pediatrics, Division of Oncology and Hematology, Charité—Universitätsmedizin Berlin, Corporate Member of Freie Universität Berlin and Humboldt-Universität zu Berlin, 13353 Berlin, Germany; 4State Institute of Forensic Medicine Berlin, 13437 Berlin, Germany; 5Connected Healthcare, Hasso Plattner Institute, University of Potsdam, 14482 Potsdam, Germany; 6Department of Dermatology, Venereology and Allergology, Charité—Universitätsmedizin Berlin, Corporate Member of Freie Universität Berlin and Humboldt-Universität zu Berlin, 10117 Berlin, Germany

**Keywords:** fibromyalgia, musculoskeletal pain, fasting, caloric restriction, nutrition, integrative medicine, complementary medicine, prolonged fasting

## Abstract

Fibromyalgia syndrome (FMS) is a common chronic pain disorder and often occurs as a concomitant disease in rheumatological diseases. Managing FMS takes a complex approach and often involves various non-pharmacological therapies. Fasting interventions have not been in the focus of research until recently, but preliminary data have shown effects on short- and medium-term pain as well as on physical and psychosomatic outcomes in different chronic pain disorders. This single-arm observational study investigated the effects of prolonged fasting (3–12 days, <600 kcal/d) embedded in a multimodal treatment setting on inpatients with FMS. Patients who were treated at the Department of Internal Medicine and Nature-Based Therapies of the Immanuel Hospital Berlin, Germany, between 02/2018 and 12/2020 answered questionnaires at hospital admission (V0) and discharge (V1), and then again three (V2), six (V3), and 12 (V4) months later. Selected routine blood and anthropometric parameters were also assessed during the inpatient stay. A total of 176 patients with FMS were included in the study. The Fibromyalgia Impact Questionnaire (FIQ) total score dropped by 13.7 ± 13.9 (*p* < 0.001) by V1, suggesting an improvement in subjective disease impact. Pain (NRS: reduction by 1.1 ± 2.5 in V1, *p* < 0.001) and quality of life (WHO-5: +4.9 ± 12.3 in V1, *p* < 0.001) improved, with a sustainable effect across follow-up visits. In contrast, mindfulness (MAAS: +0.3 ± 0.7 in V1, *p* < 0.001), anxiety (HADS-A: reduction by 2.9 ± 3.5 in V1, *p* < 0.0001), and depression (HADS-D: reduction by 2.7 ± 3.0 in V1, *p* < 0.0001) improved during inpatient treatment, without longer-lasting effects thereafter. During the study period, no serious adverse events were reported. The results suggest that patients with FMS can profit from a prolonged therapeutic fasting intervention integrated into a complex multimodal inpatient treatment in terms of quality of life, pain, and disease-specific functional parameters. ClinicalTrials.gov Identifier: NCT03785197.

## 1. Introduction

Fibromyalgia syndrome (FMS) is a musculoskeletal pain disorder which is often accompanied by comorbidities like fatigue, gastrointestinal symptomatology, poor sleep quality, and various other physical and psychological comorbidities [1,2,3]. Its prevalence is estimated from 1.8% of the world’s population to up to 8%, with females being predominantly affected [2,3,4].

The complexity of FMS is reflected in its diagnostic challenges. No radiological or laboratory markers can yet confirm its presence, and clinical symptomatology can fluctuate considerably or differ substantially between individuals. Diagnosis is usually asserted by pain scales and a history of pain in at least four of five body regions persisting for at least three months. It is not a diagnosis of exclusion, but can exist alongside other conditions, such as rheumatic diseases [4,5].

No specific causative factors have been identified or agreed upon yet, but FMS patients seem to show alterations in pain processing within the nervous system itself [4]. Further associations have been found with biosocial stressors and trauma, especially physical abuse history [6]. FMS is often accompanied by psychological symptomatology, including depression [7] and alexithymia [8].

To meet the complexity of the condition, the 2017 EULAR international therapeutic guidelines focus on multimodal treatment approaches, preferring non-pharmacological interventions over medication [9]. In a Delphi process of 2022, experts in the field supported aerobic exercise, education, sleep hygiene, and cognitive behavioral therapy as core treatments for FMS symptoms, while recommending mind–body exercises as core interventions for pain, fatigue, and sleep problems, and mindfulness for depression [10]. Clinical practice does not seem to reflect these recommendations accordingly, as overmedication has been repeatedly reported [11,12]. Up to 90% of FMS patients have been reported to consult complementary medicine as a supportive therapeutic option [3], while satisfaction with conventional standard care was rated low in a two-year cross-sectional Spanish survey [11].

Dietary interventions have not yet been incorporated into the 2017 EULAR guidelines [9], but different approaches have proven helpful in FMS treatment: in particular, studies using high-antioxidant, high-fiber, and low-processed foods as well as weight loss studies and ones applying certain nutritional supplementations have shown marked effects in alleviating FMS symptomatology [13]. Weight loss has been discussed as a separate mechanism for analgesic effects, this hypothesis being supported by preclinical models [14]. Obesity, alongside nutritional deficiencies, and consumption of food additives have also been described as possible risk factors correlated with complications in FMS symptomatology [13]. The current state of evidence suggests that dietary modifications may, at a low economic cost, improve quality of life in patients suffering from central sensitization syndromes, a complex of diseases to which FMS has been allocated [15]. Predominantly plant-based dietary patterns have shown potential in improving quality of life, sleep, pain at rest, and general health status in FMS patients [16].

Microbiome differences between patients suffering from FMS and healthy controls have been outlined in exploratory studies, but it has not yet been ascertained whether certain food components or dietary changes could positively influence FMS symptomatology in clinical settings [17,18].

Therapeutic fasting has been found to contribute to weight loss and systemic antioxidation as well as to neuroplasticity [19,20]. Additionally, improvements in mood and reduced pain perception have been observed, potentially due to increased brain availability of serotonin, endogenous opioids, and endocannabinoids [21]. Changes to the gut microbiome [22] and behavioral changes regarding dietary habits, self-efficacy, and sense of coherence and freedom [23] are being discussed, but cannot be generalized, as data are still scarce.

In the case of FMS specifically, the effect of fasting on pain perception, weight, antioxidation, and subsequent eating behavior, has thus far only been investigated in one exploratory clinical study by our research group. In a controlled, non-randomized study with 48 patients, we observed significant beneficial effects of a nature-based medical approach including fasting, compared to a conventional treatment in the same hospital [24]. Fasting is one of the main nature-based therapeutic add-ons to conventional rheumatologic treatment at the Department of Internal Medicine and Nature-Based Therapies (IMNT) at the Immanuel Hospital Berlin, Germany, where the present study was conducted. Other treatments may include physiotherapy, moderate exercise, dietary counselling, acupuncture, yoga, psychosomatic counselling, and cold or hot applications, among others. This multimodal approach renders it impossible to differentiate specific fasting effects from other therapeutic effects but presents a great opportunity to study fasting interventions under real-world conditions, more so as patients’ realities and international guidelines suggest a multimodal approach to FMS [3,9]. Furthermore, fasting in the traditional use, in German-speaking countries, has always been and still is a complex treatment, being accompanied by mind–body medicine interventions, physical exercise, relaxation, nutritional counselling, warm and cold water applications, and others, as described in respective consensus guidelines [25].

This observational study was conducted to evaluate the feasibility and effectiveness of prolonged therapeutic fasting in FMS patients in a larger cohort and with longer follow-ups than in our 2013 exploratory study.

## 2. Materials and Methods

### 2.1. Study Design

We designed this clinical study as an explorative, prospective, single-arm, single-center, open-label, observational study. Approval of the study protocol was granted by the Institutional Review Board of Charité—Universitätsmedizin Berlin (Charitéplatz 1, 10117 Berlin) in October 2015 (ID: EA4/005/17). The protocol was registered at ClinicalTrials.gov (ClinicalTrials ID: NCT03785197). It was carried out in accordance with the standards of the Declaration of Helsinki. All participants were asked for written informed consent before participating in the study. The study design has already been published elsewhere [26].

### 2.2. Setting

Recruitment took place from February 2018 to December 2020 at the Immanuel Hospital Berlin. It is a hospital of 195 beds and 5000 inpatients annually, covering orthopedic, osteological, and rheumatological cases as well as patients treated with nature-based (NB) therapies in a separate 60-bed ward. Approximately 900 patients are treated annually on the IMNT ward. The costs for most of the patients are covered by German statutory health insurance companies, while a minority of patients have private insurance or pay themselves.

The Department of IMNT at the Immanuel Hospital Berlin belongs to Europe’s leading institutions applying NB and traditional medical approaches [27,28], including traditional European medicine [29,30], on a large scale. It is especially adept at using prolonged fasting as a therapeutic measure for a variety of non-communicable diseases [31,32].

For this study, we focused on the four most common diagnoses for inpatient care in 2017, namely, FMS (ICD-code M79.7), osteoarthritis (knee and hip ICD-codes M17.9 and M16.9, respectively), rheumatoid arthritis (M06.9), and type 2 diabetes mellitus (E11.61). The results for osteoarthritis have been published before [26], while the results for the other diagnoses will be published elsewhere.

The main diagnosis for inpatient treatment was identified on admission during the first medical consultation on the IMNT ward. The individual treatment plan, including therapeutically relevant dietary interventions, was compiled accordingly by the responsible physician. If the main diagnosis was consistent with one of the abovementioned four diagnoses, and therapeutic fasting was prescribed, study personnel assessed whether patients were eligible for the study (see Section 2.4). If eligible, patients were informed about the observational study and invited to participate.

Study visits on the first or second day of hospitalization (V0) and on one of the last days before discharge (V1) involved electronic questionnaires on patient-reported outcomes that patients completed on tablets. Laboratory tests, which are part of the IMNT ward’s routine standard procedures on admission and discharge, were also reviewed for study purposes. Data on body weight, blood pressure, the use of analgesics, symptoms, and adverse effects during fasting as well as their treatment were obtained from patient files after discharge.

Throughout inpatient treatment, patients are seen by their attending physician 4 to 5 times a week, while nurses check in with each patient daily. A blood test is routinely performed on the first day after admission. Typically, a complete blood count (excluding differential blood count), blood lipids, blood glucose, electrolytes, and routine kidney and liver function parameters are included. If the findings require monitoring, the blood test is repeated before discharge. This standard procedure was introduced before the start of the study and was not changed during the entire observation period.

Follow-ups were conducted at 3-, 6-, and 12-month intervals through questionnaires. These were sent either by e-mail, if participants had provided an e-mail address, or by mail.

### 2.3. Interventions

The methodology of the fast and the multimodal therapeutic program of the Department of IMNT have already been described elsewhere in detail [26]. When the physician recommends fasting within the standard inpatient treatment, the first full day of the hospital admission is allocated as a day of preparation, involving a light plant-based diet. On this day, patients consume a calorie-restricted light diet of around 1200 kcal. Additionally, bowel cleansing is applied, facilitated by the intake of laxatives like Glauber’s salt (hydrated sodium sulfate) or through an enema. Fasting commences on the subsequent day, typically lasting a minimum of 5 days to a maximum of 12 days, whereby the exact duration depends on the patient’s constitution and inpatient stay regulations in accordance with the diagnosis, severity of illness, and ICD-10 coding. Throughout the fasting period, only water, unsweetened teas, and natural juices are consumed, resulting in a daily caloric intake of 200 to 300 calories. In the midst of the COVID-19 pandemic, the fasting regimen was switched to a fasting mimicking diet (FMD) from April 2020 until the end of the study, as there was uncertainty about the effects of fasting on the immune response to SARS-CoV-2. The total daily calorie limit was raised to maximum 600 kcal, incorporating small amounts of solid foods in the form of vegetable soup, steamed vegetables, porridge, and cooked potatoes. Fasting therapy in our department is integrated into a series of other therapeutic measures that the physician prescribed individually for each patient. Some of these nature-based therapies included nutritional counselling, cryotherapy (cold chamber), water aerobics, yoga therapy, mind–body medicine, active walking, meditation, and breathing techniques as well as psychosomatic counselling, acupuncture, physiotherapy, baths, or leech therapy. Traditionally, the fast is broken on the final day with an apple. Depending on the duration of the fasting period, solid food is gradually reintegrated over the subsequent 1 to 3 days as part of a light plant-based diet. The medication paused during the fast is gradually reintroduced on these days, if necessary.

### 2.4. Participants

All patients who were treated as inpatients in the IMNT department at Immanuel Krankenhaus Berlin in the period of February 2018 to December 2020 and who were prescribed fasting as one of the therapeutic measures by their treating physician were screened for suitability for the study.

Age between 18 and 85 years and written informed consent were further inclusion criteria.

Dementia or other severe cognitive impairments, pregnancy or breastfeeding, difficulties with the German language, and involvement in another study were exclusion criteria.

### 2.5. Variables

The feasibility and effectiveness of prolonged therapeutic fasting in FMS patients was investigated using a validated questionnaire specifically for FMS symptoms, the Fibromyalgia Impact Questionnaire (FIQ). For the FIQ, an improvement of 14 percent has been acknowledged as the minimal clinically important difference (MCID) [33]. Secondary outcomes included the following validated questionnaires: Hospital Anxiety and Depression Scale (HADS), a scale on self-efficacy (Allgemeine Selbstwirksamkeit Kurzskala, ASKU), and the Mindfulness Attention and Awareness Scale (MAAS). In addition, body weight, blood pressure, and medication as well as triglycerides, total cholesterol, LDL, and HDL were extracted from patient records.

### 2.6. Data Sources/Measurement

During the inpatient stay, electronic questionnaires were completed and, depending on the patient’s wishes, digital or analog questionnaires were used for follow-up examinations. Blood samples were taken on the day of admission and, if considered necessary by the attending physician, on one of the last days of the hospital stay. Hospital physicians took blood samples daily between 7:30 and 8:15 a.m. prior to breakfast.

### 2.7. Bias

The use of fasting as an intervention cannot be blinded for either the patient or the hospital staff. The study personnel were only responsible for recruiting participants and ensuring that questionnaires were completed. The study personnel therefore had no control over the duration of fasting, modifications of treatment modalities, or the course of the patients’ treatment.

To determine whether there was a reporting bias related to subjective improvement or worsening of symptoms in the follow-ups, patients were subdivided according to their improvements on the FIQ score at V1 (primary endpoint) into high (upper third), medium (central third), and low gainers (lower third). For the subsequent follow-ups, we cross-checked whether any of these subgroups was under- or over-represented in the answers we received.

### 2.8. Study Size

During the three-year duration of the study, an estimated 180 FMS patients were to be treated on the ward and receive fasting therapy, of which around 150 would be able to take part in the study. For an intraindividual pre–post comparison with the *t*-test and standard parameters of alpha = 0.05 (uncorrected for the number of tests, as usual in exploratory studies) and beta = 0.20 (equivalent to a power of 80%), 150 patients, including 20% dropouts, are sufficient to detect large, medium, and small effects with a minimum effect size of Cohen’s *d* ≥ 0.23.

No interim analyses were planned.

### 2.9. Statistical Methods

In this observational, exploratory, single-arm study, participants’ baseline values (V0) and vital signs were compared with those measured at subsequent visits (V1 to V4) using unadjusted *t*-tests. For the comparison between V0 and V1, all cases with complete data for that specific parameter were considered. As usual in explorative studies, no imputation was applied.

Data were analyzed using custom-written procedures in Python (v 3.9).

## 3. Results

### 3.1. Study Population

In this uncontrolled observational study, *n* = 176 patients (168 females and 8 males) suffering from FMS and following therapeutic fasting on the ward between January 2018 and December 2020 were enclosed (Figure 1). Baseline characteristics of the studied population are displayed in Table 1.

Of the originally enclosed 176 patients, 157 (90%) fasted. The fast lasted between 3 and 12 days with a mean of 7.6 (SD 1.7) (Figure 2a). The questionnaires at V1 were answered by n = 142 (82%). However, due to a failure in programming the electronic questionnaire in the early months of the study, the FIQ was not completed as a full questionnaire by the first patients, and thus shows gaps in various subscales at V1 for 21 patients (for details, see the “n” column in Table 2).

### 3.2. Questionnaires

Results revealed a significant improvement in fibromyalgia symptomatology as measured by the FIQ and its subscales (Figure 3a–e, Table 2). The total score fell from 58.3 ± 11.1 to 44.6 ± 15.5 (−23.5%) between the first and the last days of the stay, which relates to a significant reduction by 13.7 ± 13.9 points (*p* < 0.001, d = 1.02) on the FIQ scale for the total score ranging from 0 to 100, with 50 points being the average patient score. This marked reduction by 23.5 percent is larger than the minimal clinically important difference (MCID) of 14 percent. The strong improvement in the total score resulted from large effects in the subscores “Overall” (15.0 ± 4.2 to 10.9 ± 5.2, *p* < 0.0001, d = 0.87) and “Symptoms” (39.8 ± 9.0 to 30.5 ± 11.5, *p* < 0.0001, d = 0.89), a slight benefit in the FIQ subscore “Function” (3.5 ± 1.7 to 3.2 ± 2.0, *p* = 0.0328, d = 0.17), and a clinically significant effect in the pain subscore (6.8 ± 1.9 to 5.7 ± 2.6, *p* < 0.0001, d = 0.49, MCID ≥ 1 point on NRS) [34]. Effects remain on a moderate level up until V2 (after three months) for the total score and the symptoms and pain subscores and are reduced to small effects thereafter (Figure 3a–e, Table 2). There was no significant correlation between individual fasting duration in days and improvement in the FIQ score (regression: r = 0.156, *p* = 0.094).

Wellbeing as measured by the WHO-5 questionnaire (Figure 3f, Table 2) increased from 7.4 (±4.1) at baseline to 12.5 (±5.1; *p* < 0.0001) and stayed at an elevated level (>10.8 ± 5.2) until V4, again showing moderate effect sizes of d ≥ 0.5. For all questionnaire results, please refer to Table 2.

### 3.3. Physiological Parameters

The weight and blood pressure of the patients decreased during the inpatient stay, as did blood lipids. For more details on physiological, laboratory and anthropometric outcomes, please refer to Figure 2b and Table 3. Pain medication was mainly maintained or reduced, while herbal remedies for pain alleviation were given to some patients de novo. The details on the changes in medication can be found in Table 4.

### 3.4. Safety and Adverse Events

Self-reported side effects of fasting were recalled at the end of the inpatient stay or documented in the medical records, as presented in Table 5 and Figure 2d. No serious adverse events were recorded during the inpatient stay for any of the participants.

In a further analysis, it was investigated whether the positive long-term results could be explained by a selective loss of patients during the follow-up period who had not profited notably from the treatment. However, when analyzing the data of those who profited most (upper 1/3), moderately (central 1/3), and least (lower 1/3), it was found that losses to follow-up were fairly consistent across all three groups. The long-term results therefore do not appear to be biased due to a selective loss of those who benefited least or most from the intervention.

## 4. Discussion

Medically supervised fasting of max. 600 kcal daily, applied for a mean of 7–8 days as part of a multimodal inpatient nature-based intervention, showed improvements in different parameters regarding treatment of patients with FMS. These included patient-reported outcomes related to pain perception, functionality, and quality of life, along with some clinical, anthropometric, and laboratory parameters in short-, medium-, and long-term follow-ups.

One of the main symptoms of FMS is pain, accounting for frequent overmedication and overtreatment [11,12,35]. In our data set, we found a reduction in subjective and objective pain scores. The FIQ question on pain revealed a clinically significant reduction that persisted until the three-month follow-up, while patient-reported pain was still below baseline values one year after the hospital stay. Parallel to this result, the data from the patient records show that pain medication decreased slightly from baseline to discharge, while herbal remedies increased. This is especially interesting, as a current study on the use of pain medication in FMS concluded that many German patients use “on-demand” pain medication that is not in line with international guidelines, achieving only moderate pain reduction and leading to side effects [36].

Although our data are only observational in nature, the effect sizes observed (d = 1.02) compare to other nonpharmacological treatments (d = 0.63) including supportive treatments such as aerobic exercise (d = 0.89) or multidisciplinary treatment (d = 0.41) [37]. In this context, comparative effectiveness and cost-effectiveness of fasting should be further investigated.

The improvements in pain and the FIQ indicate a better sense of wellbeing. In line with this, quality of life, which was assessed using the WHO-5 questionnaire, also increased in our sample. It showed a clinically relevant increase, while depression and anxiety, measured by the HADS, decreased. Our FMS study population seems to profit more and in a shorter time frame regarding anxiety and depression than, for example, pain neuroscience education, a program usually running over several weeks [38]. The changes we observed were most pronounced from admission to discharge, but an improvement in symptomatology (FIQ subscale on symptoms), pain (FIQ subscale on pain), and mood (WHO-5) was sustained throughout the whole study period of one year.

These findings on wellbeing are of particular interest in FMS, as many patients suffer from alexithymia and depression. Four in ten FMS patients are reported to show depressive symptomatology [7]. Depression has been linked to poorer outcomes in FMS patients [7] and alexithymia has been positively associated with pain [8]. This is why interventions enhancing mindful awareness have been proposed to be integrated in therapeutic concepts in FMS [8]. In our study, we could see a rise in mindfulness, measured by the MAAS questionnaire, at V1 compared to baseline. Although this change was not sustained in further visits, the findings indicate that fasting could potentially support mindfulness practice. Fasting, being a dietary intervention perceived by many patients as drastic, and the necessity for self-restraint to adhere to it, inherently challenges self-management strategies [23]. This could be one of the mechanisms by which fasting may influence quality of life and mindfulness in FMS patients. This aspect is also important, as self-management has been discussed as a key element for treatment strategies for FMS [35].

In our study setting, fasting was not administered alone. The multimodal therapy offered entailed components that are known to improve FMS symptomatology. Evidence suggests that the promotion of moderate exercise [9,35,39], patient education [38], the introduction to mindfulness-based self-management strategies of mind–body medicine [39,40], physiotherapy [41], balneotherapy [42], manual therapies [43], and different traditional and nature-based health strategies like acupuncture, acupressure, and massage [44,45] can lead to symptom improvement. As mentioned before, fasting in the traditional sense is not only a reduction in caloric intake, but implies a complex therapeutic approach, which may include all these abovementioned interventions, depending on the clinic or medical practitioner implementing it. As such, therapeutic fasting itself could as well be described as a multimodal approach in itself.

Our findings are in line with findings from other multi-component programs for FMS. Bruce et al. have reported on a two-day multimodal treatment program embedded in routine care that yielded durable effects regarding improvement of functional status and psychological distress in a sample of 189 patients until the follow-up visit after five months [46]. Ayurvedic and conventional multimodal treatment both showed improvements on the FIQ after a two-week inpatient treatment [47]. Another multimodal program that is being developed in a community setting runs over six weeks and includes education about fibromyalgia, goal setting, pacing, sleep hygiene, and nutritional advice [48].

Other dietary interventions have also been shown to improve FMS symptomatology. A very-low-calorie diet (VLCD) of 800 kcal/d, applied in 195 obese FMS patients over three weeks, showed a minimum 30% symptom reduction in 72% of participants by week 3, while changes were not associated with the amount of weight lost [14]. The caloric restriction applied in our study was more pronounced, but shorter, and showed similar improvements. In preliminary studies from our working group, fasting with 300 kcal/d for a week ameliorated mood and wellbeing in a heterogeneous sample of chronic pain patients [49] and FIQ scores, pain, sleep, and anxiety were reduced in a prolonged fasting group compared to controls in the same hospital after two and twelve weeks. These effects could, additionally to the abovementioned mechanisms, be connected to antioxidant capacity released by caloric restriction and fasting [19]. FMS patients have been shown to produce more damaging free radicals than healthy controls, mainly affecting the nervous system, and treatments with antioxidants and vitamins have had effects on FMS symptoms [3]. In our clinical experience, fasting is often seen by patients as a potential starting point for changes in habits in general, and especially in dietary habits, as we have described formerly in a study on a religious fast [23].

In our study setting at the Department of IMNT, patients are encouraged to follow a mainly plant-based diet with low-processed foods once back at home. A review of 36 studies on nutritional approaches to fibromyalgia of 2022 concludes that low-processed foods containing high amounts of antioxidants and fibers, high-quality protein, and healthy fats showed beneficial effects in FMS [13]. A systematic review on the use of vegetarian and vegan diets in FMS published in 2021 analyzed four clinical trials and two cohort studies on the subject and concluded that following predominantly plant-based dietary patterns seems to improve biochemical parameters, quality of life, sleep, and pain at rest as well as general health status [16]. In a review of the literature on nutritional influences on central sensitization syndromes, including FMS, the authors conclude that dietary changes can considerably increase patients’ quality of life at modest costs [15]. In fact, in our clinical experience, fasting is often used as an impulse for medium- and long-term lifestyle changes, so changes in dietary habits can also be seen as part of the effects of a fasting intervention and should be tracked as such in future studies.

Several limitations restrict the generalizability of our study results. Apart from the insufficient quality of our data due to technical programming problems at the beginning of data collection, the greatest drawback of this study lies in its observational character and the lack of a control group. The inpatient setting makes randomization difficult, so that no control group could be generated on the same ward, and comparison with patients on other wards would have entailed too many differences in the multimodal therapy as to be of use in this study. Inpatients for whom fasting therapy is not indicated due to their pre-existing conditions (e.g., cachexia or eating disorder) would also not represent a suitable comparison group. In addition, fasting, like other dietary changes, cannot be blinded, which applies to both personnel and patients and further compromises the generalizability of the results. The variations in fasting length also pose a challenge to reproducibility. Additionally, the determinants of the decision-making process between the patient and the therapeutic team concerning the length of the fast has not been well documented in our study. The multimodal treatment program on the ward being individualized for every patient, and containing numerous interventions, also impedes generalizability. On the one hand, the inpatient setting alone could have unspecific positive health effects, and on the other, effects specific to the reduction in caloric intake cannot be specified due to the multimodality of the treatment. We also did not track three important therapeutic elements that merit attention: psychiatric medication, exercise, and dietary habits. Not having tracked dietary habits in the follow-ups, it is not possible to differentiate long-term fasting effects per se from health-promoting changes in dietary habits in our data. Exercise is not only critical for patients suffering from obesity, as weight loss has shown to positively influence pain in FMS, but it is the non-pharmacological treatment option with the most evidence in FMS [9]. Future studies should find ways to monitor changes in exercise, too, as less pain could contribute to more exercise and help maintain positive results after our inpatient stay.

Taking into consideration all these limitations, these data can be seen as a contribution towards the discourse concerning dietary interventions in the treatment of FMS. Our data suggest feasibility, safety, and potential advantages of medically supervised fasting for patients with FMS, when embedded in a multimodal therapeutic inpatient approach. The feasibility and safety of prolonged fasting have already been shown for various other indications, including different chronic pain syndromes [21,50,51,52].

In summary, prolonged fasting could induce multiple positive effects on symptomatology of FMS. To generate more evidence in this field, it would be commendable to study fasting in outpatient settings, as they seem easily approachable for patients suffering from FMS [48], with control groups and fewer new treatments ensuing during fasting. It would also be interesting to investigate whether effects are dose-dependent and whether fasts of shorter durations or even intermittent fasts could have similar effects. From an economic perspective, cost-effectiveness of dietary interventions, and especially fasting, should be further investigated, as fasting has been shown to lower the need for medication, sparing patients potential side effects. The rise in quality of life could also have effects on the length of sick leave. In general, if a safe and feasible intervention of 5–10 days were able to lower disease burden in FMS in the medium and long term, giving it further attention seems worthwhile. We hope that these observational data will serve as a basis for subsequent prospective interventional trials that are required to explore the concept further and ensure reproducibility of results in different settings and populations.

## 5. Conclusions

The use of prolonged modified fasting as part of a multimodal medical approach could possibly help patients with fibromyalgia regarding pain and psychosomatic symptoms.

## Figures and Tables

**Figure 1 nutrients-16-01059-f001:**
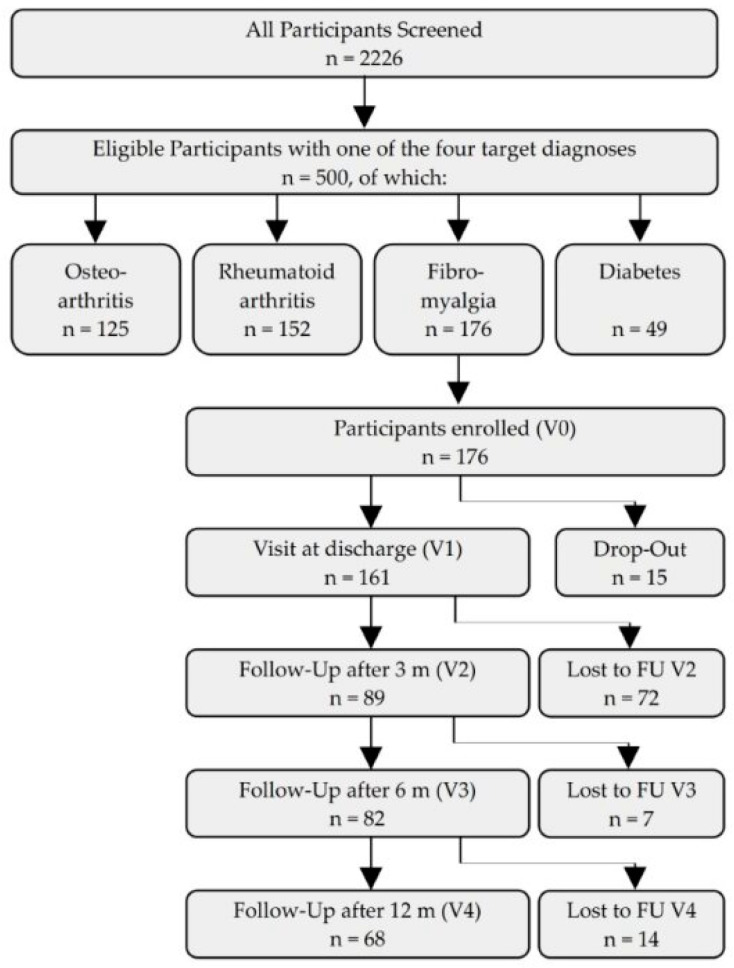
Participant flow chart. FU = follow-up, m = months, V = visit.

**Figure 2 nutrients-16-01059-f002:**
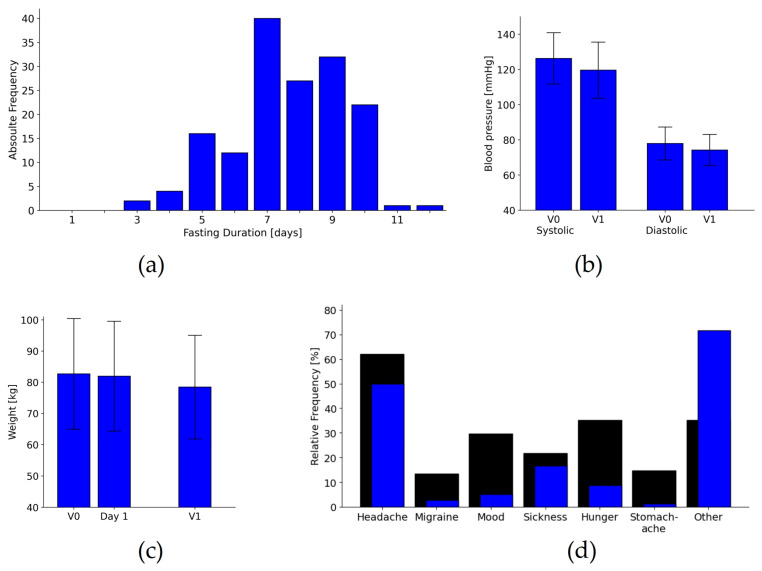
(**a**) Fasting duration histograms (in days). Results for the physiological data, blood pressure (**b**), and weight (**c**). Here, bars indicate means and SDs. (**d**) Relative frequencies of side effects of fasting. Black bars indicate self-reported frequencies in the questionnaire at discharge; blue bars show frequencies of side effects reported to the staff. V0 = Baseline visit, Day 1 = 24 h after admission, V1 = visit at discharge. “Mood” = mood disturbances.

**Figure 3 nutrients-16-01059-f003:**
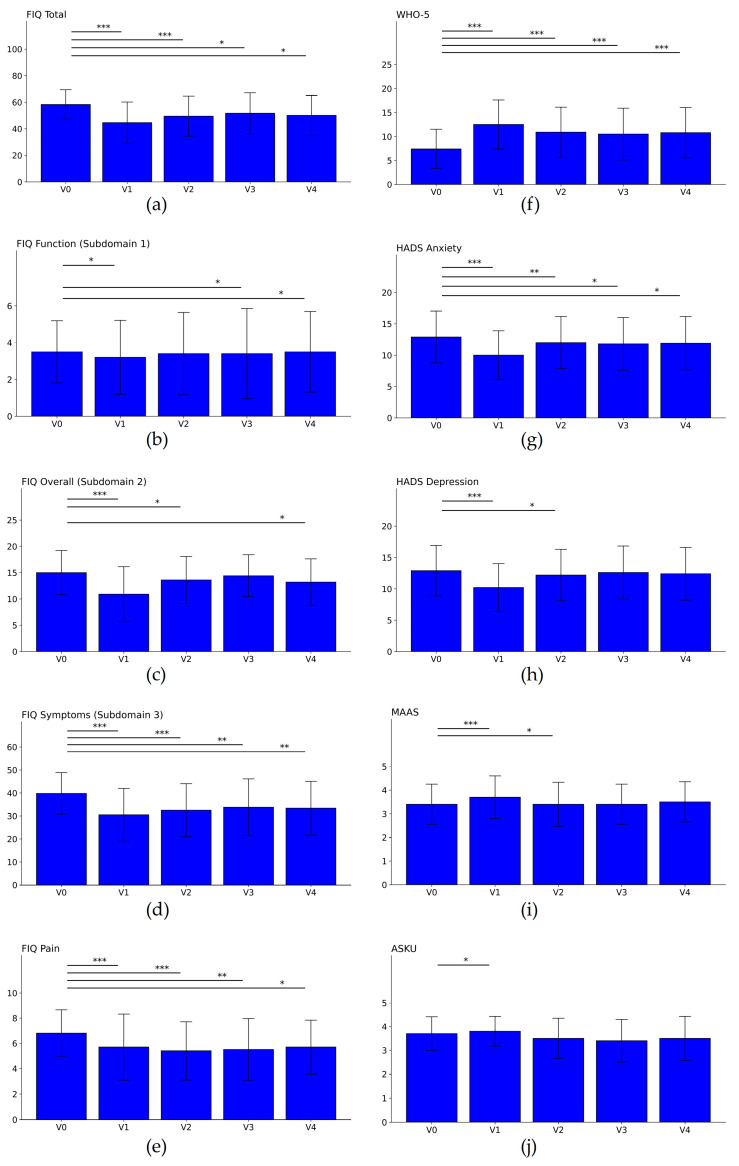
Results from the questionnaires FIQ (**a**–**e**), WHO-5 (**f**), HADS (**g**,**h**), MAAS (**i**), and (ASKU (**j**) across the five visits (V0: baseline, V1: at discharge, V2–V4: 3, 6, and 12 months after V0). Increased WHO-5 (**e**) and MAAS (**f**) scores and a decrease in all other scores suggest improvements of patients’ health. The asterisks *, **, and *** refer to *p* < 0.05, *p* < 0.01, and *p* < 0.001, respectively.

**Table 1 nutrients-16-01059-t001:** Baseline Characteristics.

Parameter	Value	Enrolled Patients
Sex	Female	168 (95.5%)
	Male	8 (4.5%)
Age [Years]	M ± SD	54.2 ± 9.8
Age Group	18–35 years	7 (4.0%)
	36–50 years	40 (22.7%)
	51–65 years	116 (65.9%)
	66–80 years	13 (7.4%)
Marital Status	Single	30 (17.0%)
	Married	107 (60.8%)
	Separated/Divorced	30 (17.0%)
	Widowed	7 (4.0%)
	Other	2 (1.1%)
Household	Single	43 (24.4%)
	With Partner	82 (46.6%)
	Single with Children	13 (7.4%)
	With Partner and Children	36 (20.5%)
	Other	2 (1.1%)
Schooling	Primary Schooling	8 (4.5%)
	Secondary Schooling	91 (51.7%)
	High School	22 (12.5%)
	University Degree	51 (29.0%)
	Other	4 (2.3%)
Occupation	Self-employed	7 (4.0%)
	Civil Servant	3 (1.7%)
	Employed	83 (47.2%)
	Worker	6 (3.4%)
	Homemaker	9 (5.1%)
	Unemployed	13 (7.4%)
	Retired	28 (15.9%)
	Permanently Disabled	20 (11.4%)
	Student	2 (1.1%)
	Other	5 (2.8%)
Gross Salary	<EUR 20,000	87 (49.4%)
	EUR 20,000–EUR 40,000	58 (33.0%)
	EUR 40,000–EUR 60,000	25 (14.2%)
	EUR 60,000–EUR 80,000	4 (2.3%)
	>EUR 80,000	2 (1.1%)
Duration of Unemployment	Up to 3 Months	71 (40.3%)
	3–6 Months	45 (25.6%)
	6 Months and Longer	12 (6.8%)
	Not unemployed	48 (27.3%)
Diabetes	Present	5 (2.8%)
Subjective Physical Health Status	Mildly Impaired	6 (3.4%)
	Impaired	98 (55.7%)
	Strongly Impaired	72 (40.9%)
Subjective Psychological Health Status	Not Impaired	10 (5.7%)
	Mildly Impaired	44 (25.0%)
	Impaired	87 (49.4%)
	Strongly Impaired	35 (19.9%)
Psychotherapy	None Thus Far	34 (19.3%)
	Earlier	90 (51.1%)
	Currently	52 (29.5%)
Nature-Based Therapies	Familiar with Concept	118 (67.0%)
Stay at This Clinic	First	102 (58.0%)
	Second	38 (21.6%)
	Third	18 (10.2%)
	Fourth	18 (10.2%)
Fasting Experience	Never	88 (50.0%)
	Once	30 (17.0%)
	Twice	22 (12.5%)
	Three times	15 (8.5%)
	Four times	6 (3.4%)
	Five or more times	15 (8.5%)
Medication	Opioids	6
	Pain Medication	161
	Neuropathic Medication	1
	Biologicals/MTX/Immuno-suppressants	4
	Prednisolone/Corticoids/ Prednisone	4
	Herbal Remedies	39
Subjective Impairment by Main Symptomatology	NRS Scale [0–10]: M ± SD	6.7 ± 1.3%
Anticipation of Inpatient Treatment Efficacy	NRS Scale [0–10]: M ± SD	6.0 ± 2.0%

Legend: M = Mean, SD = Standard Deviation, MTX = methotrexate, NRS = Numerical Rating Scale.

**Table 2 nutrients-16-01059-t002:** Questionnaire Results.

					Difference				
Parameter	Visit	M	SD	n	M	SD	T	*p*	d
FIQ Total	V0	58.3	11.07	121					
	V1	44.6	15.5	121	−13.7	13.92	10.8	<0.0001	1.02
	V2	49.2	15.47	85	−8	14.24	5.15	<0.0001	0.58
	V3	51.8	15.18	77	−5.1	15.81	2.79	0.0067	0.36
	V4	50.9	15.11	63	−4.6	14.44	2.5	0.0152	0.33
FIQ Subdomain 1 Function	V0	3.5	1.69	121					
	V1	3.2	2.01	121	−0.3	1.65	2.16	0.0328	0.17
	V2	3.7	2.36	101	0.4	2.46	1.48	0.1434	0.17
	V3	3.8	2.41	91	0.4	2.4	1.67	0.099	0.2
	V4	3.7	2.07	72	0.2	1.73	1.04	0.2998	0.11
FIQ Subdomain 2 Overall	V0	15.0	4.2	121					
	V1	10.9	5.22	121	−4.1	5.43	8.35	<0.0001	0.87
	V2	13.6	4.52	85	−0.8	5.42	1.42	0.1589	0.19
	V3	14.3	3.97	77	0.1	5.04	0.19	0.8476	0.03
	V4	13.3	4.26	63	−0.6	5.55	0.9	0.3704	0.15
FIQ Subdomain 3 Symptoms	V0	39.8	9.03	121					
	V1	30.5	11.48	121	−9.3	10.46	9.54	<0.0001	0.89
	V2	33.9	12.51	101	−5.9	10.92	5.4	<0.0001	0.53
	V3	35.2	12.14	91	−4.2	12.62	3.19	0.002	0.39
	V4	35.1	11.74	72	−3.3	11.21	2.5	0.0148	0.31
FIQ Pain (NRS)	V0	6.8	1.86	121					
	V1	5.7	2.62	121	−1.1	2.54	4.85	<0.0001	0.49
	V2	5.7	2.49	101	−1	2.44	4.27	<0.0001	0.47
	V3	5.9	2.53	91	−0.7	2.35	2.75	0.0072	0.31
	V4	6.1	2.18	72	−0.6	2.14	2.41	0.0186	0.29
WHO-5	V0	7.3	4.16	176					
	V1	12.5	5.12	142	5	4.87	12.25	<0.0001	1.08
	V2	11.1	5.28	101	3.6	5.47	6.66	<0.0001	0.75
	V3	10.5	5.29	91	2.5	5.86	4.08	0.0001	0.51
	V4	10.8	5.05	72	2.6	5.21	4.14	0.0001	0.53
MAAS	V0	3.4	0.85	176					
	V1	3.7	0.9	142	0.3	0.72	5.46	<0.0001	0.38
	V2	3.4	0.94	101	0	0.75	0.55	0.5825	0.05
	V3	3.4	0.86	91	0	0.84	0.36	0.7184	0.04
	V4	3.5	0.87	72	0	0.82	0.04	0.966	0
HADS Depression	V0	13	4.07	176					
	V1	10.2	3.8	142	−2.7	2.98	10.66	<0.0001	0.68
	V2	12.1	4.12	101	−0.8	3.68	2.15	0.0338	0.2
	V3	12.6	4.23	91	0	4.03	0.05	0.9588	0.01
	V4	12.3	4.09	72	−0.2	4.36	0.32	0.7481	0.04
HADS Anxiety	V0	12.9	4.02	176					
	V1	10	3.88	142	−2.9	3.53	9.82	<0.0001	0.73
	V2	12	4.12	101	−1	3.16	3.04	0.003	0.23
	V3	11.8	4.13	91	−0.8	3.08	2.5	0.0141	0.2
	V4	11.9	4.17	72	−1	3.78	2.32	0.0231	0.25
ASKU	V0	3.7	0.7	176					
	V1	3.8	0.63	142	0.1	0.57	1.8	0.0746	0.13
	V2	3.5	0.85	101	−0.2	0.86	2.82	0.0057	0.31
	V3	3.4	0.89	91	−0.4	0.86	4.5	<0.0001	0.52
	V4	3.5	0.9	72	−0.3	0.83	2.72	0.0083	0.32

Legend: The left-hand side shows descriptive statistics for each visit separately, while the right-hand side shows the mean of individual differences between the baseline visit (V0) and the respective visit (V1: at discharge, V2–V4: 3, 6, and 12 months after V0). Differences and statistics have been calculated only for complete cases for the individual parameter and visit. FIQ = Fibromyalgia Impact Questionnaire, HADS = Hospital Anxiety and Depression Scale, PSS = Perceived Stress Scale, WHO-5 = Quality of Life, ASKU = Allgemeine Selbstwirksamkeit Kurzskala (self-efficacy scale), MAAS = Mindfulness Attention Awareness Scale, M = Mean, SD = Standard Deviation, n = number of participants, T = test statistic, *p* = *p*-value of the paired *t*-test, d = Effect size (Cohen’s *d*).

**Table 3 nutrients-16-01059-t003:** Results of Physiological and Laboratory Parameters.

					Differences				
Parameter	Visit	M	SD	n	M	SD	T	*p*	d
Cholesterol [mg/dL]	V0	237.2	43.75	102					
	V1	202	50.1	79	−36.9	35.28	9.23	<0.0001	0.81
LDL [mg/dL]	V0	150.9	38.42	97					
	V1	133	41.45	70	−20.5	30.08	5.67	<0.0001	0.53
HDL [mg/dL]	V0	62.6	13.31	97					
	V1	52.2	12.24	70	−11.7	8.58	11.35	<0.0001	0.91
Triglycerides [mg/dL]	V0	119.6	51.32	102					
	V1	106.5	26.51	78	−11.5	42.26	2.39	0.0194	0.28
Weight [kg]	V0	82.7	17.72	161					
	Day 1	81.9	17.56	161					
	V1	78.5	16.56	161	−3.5	1.97	22.36	<0.0001	0.2
Systolic BP [mmHg]	V0	126.3	14.56	161					
	V1	119.6	15.9	161	−6.7	17.3	4.88	<0.0001	0.44
Diastolic BP [mmHg]	V0	77.9	9.33	161					
	V1	74.3	8.82	161	−3.7	9.92	4.67	<0.0001	0.4

Legend: The left-hand side shows descriptive statistics for each visit separately, while the right-hand side shows the differences between the baseline visit (V0) and V1 (visit at discharge). As patients were received at the hospital after breakfast, weight was determined on the morning after admission (day 01) and compared with V1. M = Mean, SD = Standard Deviation, n = number of participants, T = test statistic, *p* = *p*-value of the paired *t*-test, d = Effect size (Cohen’s d).

**Table 4 nutrients-16-01059-t004:** Changes in Medication.

	Discontinued	Reduced	Reduced to On-Demand	Unchanged	Increased to On-Demand	Increased
Opioids	0	2	0	3	0	1
NSAIDs	8	31	1	112	0	9
Neuropathic Medication	0	1	0	0	0	0
Biologicals/MTX/Immunosuppressants	0	0	0	3	0	1
Prednisolone/Corticoids/ Lodotra	0	2	0	2	0	0
Herbal Remedies	1	0	1	1	20	16

Legend: Pain medication was categorized in opioids/opiates and non-steroidal anti-inflammatory drugs (NSAIDs), including Ibuprofen, Diclofenac, Coxibs, Paracetamol, and Metamizole among others. Medications used to treat neuropathic pain (such as Carbamazepine, Gabapentin), Biologicals, Methotrexate (MTX) or Corticoids as well as herbal remedies were reported separately. We developed a scale for the changes in medication except herbal remedies: −2 (medication was discontinued), −1 (medication was significantly reduced, including discontinuation of rescue medication or reduction from daily use to rescue medication), 0 (no substantial change in dosage), +1 (new medication or 1.5 to 2.5-fold rise in dosage), and +2 (at least 3-fold increase in medication). For herbal medicines, we rated −2 as stopping herbal medicines taken at admission, −1 as a notable dose reduction, −0.5 change from daily intake of herbal medicines to rescue medication, 0 as no substantial change, +0.5 as a slight increase in dosage, including, i.e., new herbal rescue medication, +1 as a new daily intake of herbal medicines, +1.5 a new daily herbal medicine plus a herbal medicine as rescue medication, and +2 as two new daily herbal remedies.

**Table 5 nutrients-16-01059-t005:** Subjectively Reported Side Effects of Fasting.

Side Effect	Data from Recall Questionnaire	Extracted from Patient Records
Headache	88 (62.0%)	69 (50.0%)
Migraine	19 (13.4%)	4 (2.9%)
Mood Disturbance	42 (29.6%)	7 (5.1%)
Sickness	31 (21.8%)	23 (16.7%)
Hunger	50 (35.2%)	12 (8.7%)
Stomachache	21 (14.8%)	2 (1.4%)
Other	50 (35.2%)	99 (71.7%)

Values are reported in absolute and relative frequencies (%).

## Data Availability

The data are not publicly available due to privacy issues (patient data). Data can be obtained from the corresponding author on justified request.

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
