# Peer review of "Effects of Prolonged Medical Fasting during an Inpatient, Multimodal, Nature-Based Treatment on Pain, Physical Function, and Psychometric Parameters in Patients with Fibromyalgia: An Observational Study"

_nutrients, 2024, doi:10.3390/nu16071059_

Round 1

Reviewer 1 Report

Comments and Suggestions for Authors

The present study examined the effects of prolonged fasting embedded in a multimodal treatment setting on inpatients with Fibromyalgia syndrome. The results highlighted that patients with Fibromyalgia syndrome may profit from a prolonged therapeutic fasting intervention embedded in a complex multimodal inpatient treatment regarding quality of life, pain, and disease-specific functional parameters.

This is a well-written paper. The manuscript is well structured, with the primary purpose so clear. Methods are adequate, and the data are clearly shown. The data support conclusions. All the factors mentioned above are described clearly. 

A significant limitation of the study is the lack of a control group. This makes it impossible to assess the reliability of the results thoroughly. In the limitations section, the authors describe the impossibility of using a control group in this experiment. In the future, it would be worth programming the study differently to compare the medical fasting method with the control group. 

In general, the work is exciting and can contribute to the literature. 

Author Response

A significant limitation of the study is the lack of a control group. This makes it impossible to assess the reliability of the results thoroughly. In the limitations section, the authors describe the impossibility of using a control group in this experiment. In the future, it would be worth programming the study differently to compare the medical fasting method with the control group.

  • Answer: Thank you for pointing this major limitation out. We agree that it would be worth planning a study with a control group in future and hope to accomplish this in the near future. Thank you for your encouraging comments!

Reviewer 2 Report

Comments and Suggestions for Authors

Thank you very much for giving me the opportunity to review presented manuscript which, in an extended paper, examines the impact of prolonged medical fasting on pain, physical function, and psychometric parameters in patients with fibromyalgia. The study is suffciently documented and well written as well as I appreciate that the limitations of the study are very well presented. Figures and tables are readable and clearly presented. Discussion is a strong point of the manuscript. I have the comment regarding bibiography. Below I present a few of my suggestion

1.       In one sentence “The Department of IMNT at the Immanuel Hospital Berlin belongs to Europe´s leading institutions applying NB and traditional medical approaches [27-32] including tradi tional European medicine [33-39] on a large scale. It is especially adept at using prolonged 138 fasting as a therapeutic measure for a variety of non-communicable diseases [21, 22, 24, 139 25, 40-59]” the authors cited 6 + 7 + 5+ 20 = 38 articles to confirm that The Department of IMNT at the Immanuel Hospital Berlin belongs to Europe´s leading institutions. The aim of the paper is not to appreciate this department so in my opinion 1-2 citations could refer to this fact while in article almost 50% is engaged for this purpose. Please reduce significantly these redundant references.

2.       As I mentioned earlier I appreciate that in the limitations Authors mentioned about: technical programming problems, the lack of a control group, the variations in fasting length which impede generalizability of the results and drawing univocal causative conclusions. Authors also noted that not having tracked dietary habits in the follow-ups, unabled to differentiate long-term fasting effects per se from health promoting changes in dietary habits in the data. In my opinion this is very important limitation and should be placed srictly in paragraph with limitations (instead of preceding it).

3.       In Figure 3 b FIQ Function subdomain 1 please change upper limit of y-axis to 6 instead of 10 – in a current form it seems that all bars has exactly the same hight while significant differences were detected. Please increase readability of this panel of the plot.

4.       In Table 1 some notation is misleading e.g. – column is titled “Parameter” and it contains the first category “Total”. The name of the last column “All Patients” is also misleading. In Age row m+/-SD is presented in the form 54.2 (+/-9.8). Please unify to M+/-SD and 54.2+/-9.8 and add unit “Years” to each age category. Similarly for other variables. Explain N/A – not available?

5.       Please add legend under Table 1 (similarly as you did under Table 2)

6.       Under Table 5 it is written that “Values are reported n (%) unless specified otherwise” – is necessary to add “unless specified otherwise”

Author Response

Thank you very much for giving me the opportunity to review presented manuscript which, in an extended paper, examines the impact of prolonged medical fasting on pain, physical function, and psychometric parameters in patients with fibromyalgia. The study is suffciently documented and well written as well as I appreciate that the limitations of the study are very well presented. Figures and tables are readable and clearly presented. Discussion is a strong point of the manuscript. I have the comment regarding bibiography. Below I present a few of my suggestion

  1. In one sentence “The Department of IMNT at the Immanuel Hospital Berlin belongs to Europe´s leading institutions applying NB and traditional medical approaches [27-32] including tradi tional European medicine [33-39] on a large scale. It is especially adept at using prolonged 138 fasting as a therapeutic measure for a variety of non-communicable diseases [21, 22, 24, 139 25, 40-59]” the authors cited 6 + 7 + 5+ 20 = 38 articles to confirm that The Department of IMNT at the Immanuel Hospital Berlin belongs to Europe´s leading institutions. The aim of the paper is not to appreciate this department so in my opinion 1-2 citations could refer to this fact while in article almost 50% is engaged for this purpose. Please reduce significantly these redundant references.

- Answer: Thank you for looking into our citations. It is very helpful to hear that fewer citations would also suffice to show the team’s experience in the field. We have reduced the citations to two for every point. Do you feel the selection is representative and clear?

  1. As I mentioned earlier I appreciate that in the limitations Authors mentioned about: technical programming problems, the lack of a control group, the variations in fasting length which impede generalizability of the results and drawing univocal causative conclusions. Authors also noted that not having tracked dietary habits in the follow-ups, unabled to differentiate long-term fasting effects per se from health promoting changes in dietary habits in the data. In my opinion this is very important limitation and should be placed srictly in paragraph with limitations (instead of preceding it).

- Answer: Thank you for pointing this out. We have transferred that sentence to the limiations section and concluded the paragraph before differently.

  1. In Figure 3 b FIQ Function subdomain 1 please change upper limit of y-axis to 6 instead of 10 – in a current form it seems that all bars has exactly the same hight while significant differences were detected. Please increase readability of this panel of the plot.
  2. In Table 1 some notation is misleading e.g. – column is titled “Parameter” and it contains the first category “Total”. The name of the last column “All Patients” is also misleading. In Age row m+/-SD is presented in the form 54.2 (+/-9.8). Please unify to M+/-SD and 54.2+/-9.8 and add unit “Years” to each age category. Similarly for other variables. Explain N/A – not available?
  3. Please add legend under Table 1 (similarly as you did under Table 2)
  4. Under Table 5 it is written that “Values are reported n (%) unless specified otherwise” – is necessary to add “unless specified otherwise”

- Answer: Thank you for reading our manuscript to carefully. We have adapted the points in Figure 3, Table 1 and Table 5.

Reviewer 3 Report

Comments and Suggestions for Authors

Koppold et al., through an observational study built on previous researches of the same team, sought to investigate whether prolonged fasting for 12 months, integrated in a multimodal intervention, was able to improve the symptoms of a cohort of patients with fibromyalgia. Based on the results obtained, it is concluded that this fasting regimen significantly improved pain and quality of life. The study design has a number of limitations which were duly acknowledged in the discussion.

Language is excellent and the figures are acceptable. The protocol was registered at a repository.

I have some concerns mainly related to data presentation.

Page 5, line 208. If the minimal clinically important difference had to be at least 14%, then some of the statistically significant differences reported in Table 1 did not reach this threshold.

Page 11, Table 2. Given that the statistical comparisons ‘before-after’ were carried out using a t-test for paired data, I don't understand why in the Table 2 differences between the groups means were reported and not, instead, the mean of single differences.

Page 13, lines 343-349. The authors admitted that for some parameters a certain number of participants were lost between V0 and V1, suggesting that some statistically significant improvements were actually due to the drop out of subjects with the worst score. Is anything known about patients lost to follow-up?

Page 15, lines 422-424. The reference no. 19 did not mention only the antioxidant capacity but also the increased autophagy and recycling at the cellular level.

Minor remarks

Page 3, line 142. In my opinion the wording "ICD-code" should also be reported in the case of M79.7

Page 4, line 176. I'm not sure, but Glauber's salt is a more widespread term in German literature for hydrated sodium sulfate.

Page 7, Table 1. The last percentage of the 'Household' row could be adjusted from 1.2% to 1.1% to make the sum equal to 100.

Author Response

Page 5, line 208. If the minimal clinically important difference had to be at least 14%, then some of the statistically significant differences reported in Table 1 did not reach this threshold.

  • Answer: The minimally clinically important difference (MID) of 14% only applies to the Fibromyalgia Impact Questionnaire, as every questionnaire and laboratory parameter has different levels of clinical importance. As MIDs are usually reported for the main outcome parameter, we have done so in our manuscript, too.

Page 11, Table 2. Given that the statistical comparisons ‘before-after’ were carried out using a t-test for paired data, I don't understand why in the Table 2 differences between the groups means were reported and not, instead, the mean of single differences.

  • Answer: Thank you for mentioning this point. In Table 2 we have actually reported the mean of individual differences, as we also find this the more accurate way of reporting statistical data. We have adapted the legend to be more clear, so it now reads as follows: “The left-hand side shows descriptive statistics for each visit separately, while the right-hand side shows the mean of individual differences.”

Page 13, lines 343-349. The authors admitted that for some parameters a certain number of participants were lost between V0 and V1, suggesting that some statistically significant improvements were actually due to the drop out of subjects with the worst score. Is anything known about patients lost to follow-up?

  • Answer: Yes, this is a very important point. In our methods section we have described the following (2.7 Bias): “We divided patients into high (top third), medium (middle third) and low (bottom third) gainers according to their improvements in FIQ score at V1 (primary endpoint) to evaluate if there was any reporting bias in terms of perceived improvement or worsening of symptoms at follow-up. Subsequent follow-ups examined whether any of these subgroups were under- or over-represented in the responses received.”, and in our results section we have described: “In a further analysis, it was investigated whether the positive long-term results could be explained by a selective loss of patients during the follow-up period who had not profited notably from the treatment. But when analyzing the data of those who profited most (upper 1/3), moderately (central 1/3), and least (lower 1/3) it was found that losses to follow-up were fairly consistent across all three groups. The long-term results therefore do not appear to be biased due to a selective loss of those who benefited least or most from the intervention.” We hope this clarifies the point.

Page 15, lines 422-424. The reference no. 19 did not mention only the antioxidant capacity but also the increased autophagy and recycling at the cellular level.

Answer: Yes, this reference is one really rich in information. Do you feel we should add autophagy to the manuscripts introduction? We are not aware of a direct link between autophagy and fibromyalgia pathology, this is why we had not included it until now.

Minor remarks

Page 3, line 142. In my opinion the wording "ICD-code" should also be reported in the case of M79.7

  • Answer: Thank you for pointing this out, we have added this information in the manuscript.

Page 4, line 176. I'm not sure, but Glauber's salt is a more widespread term in German literature for hydrated sodium sulfate.

  • Answer: Thank you for pointing this out, we have added this information to our manuscript.

Page 7, Table 1. The last percentage of the 'Household' row could be adjusted from 1.2% to 1.1% to make the sum equal to 100.

  • Answer: Thank you, we have adapted the number in Table 1 accordingly.

Round 2

Reviewer 2 Report

Comments and Suggestions for Authors

In the current version, the Authors improved their manuscript and apply all my suggestions and comments. In particular, they significantly reduced the citations - which definitely were redundant. In this version I can accept the manuscript.

Reviewer 3 Report

Comments and Suggestions for Authors

The authors responded satisfactorily to my concerns. In particular I appreciated the explanations provided in the case of patients lost to follow-up. As for mentioning autophagy in the introduction, I don't think it's necessary, I just wanted to draw the authors' attention to the information contained in the relative reference. I think the manuscript is much improved, and I have nothing more to ask.